# MulVuln: Enhancing Pre-trained LLMs with Shared and Language-Specific Knowledge for Multilingual Vulnerability Detection

## Abstract

Software vulnerabilities (SVs) pose a critical threat to safety-critical systems, driving the adoption of AI-based approaches such as machine learning and deep learning for software vulnerability detection. Despite promising results, most existing methods are limited to a single programming language. This is problematic given the multilingual nature of modern software, which is often complex and written in multiple languages. Current approaches often face challenges in capturing both shared and language-specific knowledge of source code, which can limit their performance on diverse programming languages and real-world codebases. To address this gap, we propose MulVuln, a novel multilingual vulnerability detection approach that learns from source code across multiple languages. MulVuln captures both the shared knowledge that generalizes across languages and the language-specific knowledge that reflects unique coding conventions. By integrating these aspects, it achieves more robust and effective detection of vulnerabilities in real-world multilingual software systems. The rigorous and extensive experiments on the real-world and diverse REEF dataset, consisting of 4,466 CVEs with 30,987 patches across seven programming languages, demonstrate the superiority of MulVuln over thirteen effective and state-of-the-art baselines. Notably, MulVuln achieves substantially higher F1-score, with improvements ranging from 1.45% to 23.59% compared to the baseline methods.

## 1 Introduction

Software vulnerabilities (SVs) are flaws or oversights in programs that attackers can exploit to compromise systems, manipulate sensitive data, or disrupt operations (Dowd et al., 2006; Fu et al., 2024b). Due to the widespread use of software, such vulnerabilities pose significant security risks. The increasing severity and impact of SVs have driven the development of automated techniques capable of efficiently detecting vulnerabilities with minimal human intervention (Li et al., 2016; 2018b; Nguyen et al., 2019; 2020; Ding et al., 2022; Fu et al., 2024a; Nguyen et al., 2025).

Detecting SVs is essential to ensure the security and reliability of software applications (Dowd et al., 2006; Lin et al., 2020; Hanif et al., 2021; Nguyen et al., 2021; Liu et al., 2023; Fu et al., 2023c). Identifying vulnerable programs or functions enables security teams to prioritize resources and address critical issues during software development and testing. To support this, a variety of SVD systems have been developed, ranging from open-source to commercial tools and from manual to fully automated approaches (Neuhaus et al., 2007; Shin et al., 2011; Grieco et al., 2016; Li et al., 2018b; Duan et al., 2019; Cheng et al., 2019; Wattanakriengkrai et al., 2020; Nguyen et al., 2024).

Most prior work in software vulnerability detection (SVD) relied on handcrafted features manually designed by domain experts (Yamaguchi et al., 2011; Shin et al., 2011; Grieco et al., 2016; Kim et al., 2017). Such features can be outdated, biased, and often fail to generalize across projects (Zimmermann et al., 2009). To overcome these limitations, deep learning-based approaches have been developed to automatically learn features from source code, demonstrating superior performance compared to manual feature engineering (Dam et al., 2018; Li et al., 2018a; Fu et al., 2023b; 2024c; Nguyen et al., 2024; 2025). More recently, both code-specific pre-trained language models (PLMs, e.g., CodeBERT (Feng et al., 2020) and CodeT5 (Wang et al., 2021)) and large language

models (LLMs), including code-specialized models (e.g., CodeLlama (Rozière et al., 2024)) and general-purpose models (e.g., ChatGPT (OpenAI, 2022)), have been increasingly explored for software vulnerability detection (Gao et al., 2023; Fu et al., 2023c; Yao et al., 2024). These studies highlight the promising capability of such models to extract fundamental knowledge (i.e., general patterns) from source code, thereby facilitating effective vulnerability detection.

Although machine learning, deep learning, and large and pre-trained language model-based approaches have advanced vulnerability detection, most of them are limited to a single programming language, typically C or C++, using datasets such as CVEfixes (Bhandari et al., 2021) and Big-Vul (Fan et al., 2020). This limitation reduces their practical applicability, as real-world software projects are increasingly complex, often involving multiple languages such as Python and Go (Al-fadel et al., 2023; Hu et al., 2024; Li et al., 2022), and vulnerabilities exist across these diverse ecosystems. Many applications are polyglot, containing components in multiple languages (Li et al., 2022), and even non-C/C++ projects can harbor serious vulnerabilities with potentially catastrophic consequences (Livshits & Lam, 2005; Alfadel et al., 2023; Mussbacher et al., 2024). Therefore, models restricted to a single language struggle to generalize and have limited use in contemporary software development, highlighting the need for multilingual vulnerability detection approaches.

To address this, we propose MULVULN, a novel approach to multilingual vulnerability detection. MULVULN is designed to capture both shared knowledge (enhancing generalization and transferability across programming languages) and language-specific knowledge (reflecting the unique characteristics of each language and allowing the model to adapt more effectively). By jointly leveraging these two capabilities, our proposed MULVULN approach is designed to enable more robust and effective multilingual vulnerability detection. Specifically, MULVULN consists of two main parts. The first leverages a PLM to capture shared knowledge across languages and encode essential semantic and syntactic relationships crucial for vulnerability detection. The second introduces a parameter pool to model language-specific features, allowing the model to adapt to the unique characteristics of each programming language. Together, these parts form a unified framework for solving the multilingual vulnerability detection problem.

In summary, our key contributions are as follows:

- We study the important problem of multilingual vulnerability detection, a research area where automated AI-based approaches remain relatively underexplored.

- We propose MULVULN, an innovative deep learning-based approach for solving the problem. MULVULN leverages a PLM to capture shared cross-language knowledge and encode semantic and syntactic patterns, providing generalization ability across diverse programming languages. In addition, we introduce a parameter pool to model language-specific features, enabling the model to adapt to unique characteristics of each language. Together, these capabilities lead to more robust and effective multilingual vulnerability detection. To the best of our knowledge, our work is among emerging approaches proposed to address the problem and can serve as a strong baseline for future research.

- We evaluate our MULVULN approach on the real-world and diverse multilingual source code REEF dataset, consisting of 4,466 CVEs with 30,987 patches across seven programming languages (i.e., C, C++, C#, Go, Java, JavaScript, and Python). Rigorous experiments demonstrate the effectiveness and superiority of our approach over thirteen effective, state-of-the-art vulnerability detection baselines in the multilingual setting.

## 2 RELATED WORK

AI-based approaches have been extensively explored for software vulnerability detection (SVD), ranging from handcrafted features manually designed by domain experts (Yamaguchi et al., 2011; Shin et al., 2011; Li et al., 2016; Grieco et al., 2016; Kim et al., 2017) to automatic feature learning using deep learning–based methods (Li et al., 2018b; Lin et al., 2018; Dam et al., 2018; Li et al., 2018a; Duan et al., 2019; Cheng et al., 2019; Zhuang et al., 2020; Nguyen et al., 2022; Fu et al., 2023a; Nguyen et al., 2024). For example, Dam et al. (2018) employed a deep neural network to convert sequences of code tokens into vector representations, which were then fed into a separate classifier, whereas Li et al. (2018b) jointly learned the vector representation and trained the classifier within a single deep network. Advanced deep learning architectures have further been investigated

for addressing the SVD problem. Russell et al. (2018) combined recurrent neural networks (RNNs) and convolutional neural networks (CNNs) to extract features from embedded source code representations, while Zhuang et al. (2020); Nguyen et al. (2022); Cao et al. (2024) proposed graph neural network (GNN)-based models, TMP, ReGVD, and Coca, respectively, for SVD.

Recent studies have investigated large language models (LLMs) and pre-trained language models (PLMs) for vulnerability detection (Feng et al., 2020; Guo et al., 2021; Wang et al., 2021; Gao et al., 2023; Fu et al., 2023c; Yao et al., 2024; Bahaa et al., 2024; Liu et al., 2024). PLMs such as CodeBERT, GraphCodeBERT, and CodeT5 support multiple programming languages and tasks including code search, completion, and summarization (Feng et al., 2020; Guo et al., 2021; Wang et al., 2021). Fine-tuning these models for downstream tasks like SVD has shown promising results. Recent work (Gao et al., 2023; Fu et al., 2023c; Yao et al., 2024; Yin et al., 2024; Lu et al., 2024) has evaluated LLMs such as ChatGPT and CodeLlama on SVD, demonstrating their potential while also revealing limitations due to the lack of explanatory context in downstream datasets and the complexity of the task. These studies suggest that providing additional context beyond the source code may help LLMs better capture code intricacies and improve vulnerability predictions.

Large language models (LLMs), including code-specialized and general-purpose models, as well as code-specific pre-trained language models (PLMs) have recently been investigated and shown potential for multilingual vulnerability detection downstream task via fine-tuning or prompt engineering (Shu et al., 2025), as their pre-training on large-scale, diverse codebases enables them to capture general patterns and knowledge across multiple programming languages. However, when applied to downstream tasks such as multilingual vulnerability detection, these models often struggle to capture fine-grained distinctions and language-specific characteristics, which can limit their effectiveness in accurately identifying vulnerabilities.

## 3 THE PROPOSED MULVULN APPROACH

### 3.1 PROBLEM STATEMENT

We denote $\mathcal{D}$ as a real-world multilingual source code dataset across multiple programming languages (e.g., C, C++, Java, Python, and JavaScript), consisting of $\{(X_1, Y_1), \ldots, (X_N, Y_N)\}$, where $X_i$ is a source code sample (i.e., a function) and $Y_i \in \{0, 1\}$ is its vulnerability label (0: non-vulnerable, 1: vulnerable). In this paper, we study the problem of multilingual vulnerability detection, which aims to automatically predict the label $Y_i$ for each source code sample $X_i$.

### 3.2 METHODOLOGY

In what follows, we present the details of how our MULVULN approach works and addresses the multilingual vulnerability detection problem. The first part of MULVULN leverages a pre-trained language model (PLM) to capture shared knowledge across languages, encoding both semantic and syntactic relationships essential for robust vulnerability detection, enhancing generalization and transferability across programming languages. The second part introduces a parameter pool to model language-specific characteristics, allowing the model to adapt to the unique features of each programming language. Together, these parts form a unified framework that aims to enhance robustness and effectiveness in solving the multilingual vulnerability detection problem. An overall visualization is depicted in Figure 1.

### 3.2.1 SHARED KNOWLEDGE LEARNING WITH PRE-TRAINED LANGUAGE MODELS

Pre-trained language models (PLMs) (e.g., CodeT5 (Wang et al., 2021)) are trained on large-scale source code datasets covering diverse programming languages. They have demonstrated excellent performance on various downstream software engineering tasks, including code summarization, code search, and vulnerability detection. More importantly, PLMs have the capability to learn shared knowledge by capturing generalizable semantic and syntactic patterns across programming languages. This shared knowledge provides a foundation for multilingual vulnerability detection by supporting cross-language generalization and robust feature representations (Shu et al., 2025).

Inspired by this capability of PLMs, as illustrated in Figure 1, the primary part of our proposed MULVULN approach leverages a PLM (e.g., the encoder of CodeT5) to capture shared knowledge that generalizes across languages and encodes essential semantic and syntactic relationships from source code, supporting multilingual vulnerability detection.

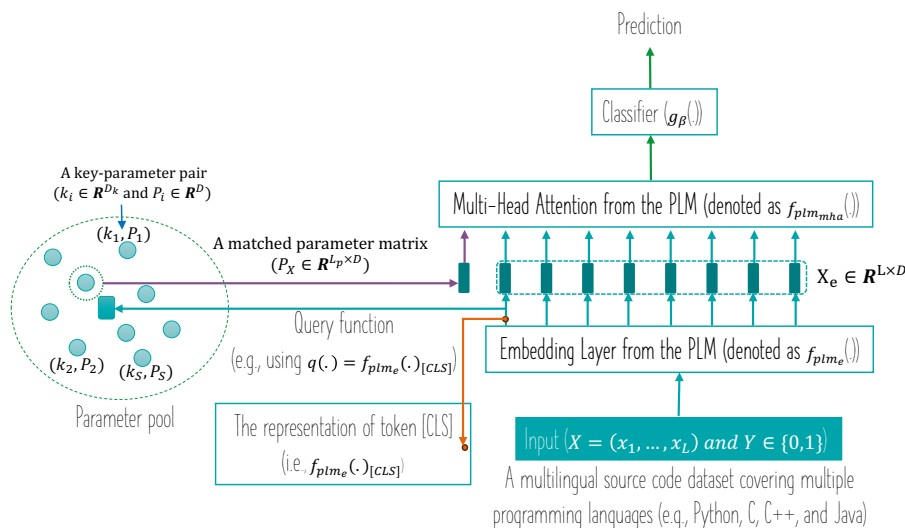

Figure 1: Overview of MULVULN for multilingual vulnerability detection by enhancing a PLM (e.g., the encoder of CodeT5 including $f_{plm_e}(.)$ and $f_{plm_{mha}}(.)$) with shared and language-specific knowledge. For each input $X$, basically, a single parameter matrix $P_X \in \mathbf{R}^{L_p \times D}$ is selected from the parameter pool $\mathcal{P}$ to form the adapted input embedding $X_p = \mathrm{concat}(P_X, X_e)$, encoding both shared and language-specific information. By default, we use the *[CLS]* token representation for the query function, and the classifier input aggregates the multi-head attention outputs corresponding to the tokens in the selected parameter matrix using mean pooling.

### 3.2.2 PARAMETER POOL FOR LANGUAGE-SPECIFIC KNOWLEDGE

Despite PLMs excelling at learning general patterns across multiple programming languages due to large-scale pretraining, they remain limited in fully capturing and clearly distinguishing language-specific nuances, such as subtle syntax rules, idiomatic coding patterns, or unique conventions of each programming language (Lu et al., 2021; Cassano et al., 2023; Du et al., 2024). This limitation becomes particularly important in downstream tasks like multilingual vulnerability detection, where fine-grained distinctions and language-specific characteristics across languages are essential for accurate detection.

To mitigate this problem, we propose a parameter pool containing additional parameters specifically designed to encode fine-grained distinctions and language-specific characteristics of each programming language. For each input source code sample from a particular language, we implement a key–parameter pair-based query mechanism that allows the model to dynamically select the most suitable parameter. The selected parameter is then concatenated with the input embeddings to form the input for the PLM, enabling the model to capture shared knowledge while preserving language-specific distinctions, thereby supporting more robust learning and accurate prediction.

The parameter pool is designed to encode the distinct knowledge of each programming language from its corresponding source code inputs. Formally, the parameter pool is defined as: $\mathcal{P} = \{P_1, P_2, \ldots, P_S\}$, where $S$ is the total number of parameters. By default, $S$ is set equal to the number of programming languages, with each language encouraged to use its own corresponding parameter $P_j$. Note that each $P_j \in \mathbf{R}^{L_p \times D}$ denotes a parameter matrix of length $L_p$ and embedding size $D$, consistent with the embedding dimension of the source code token embeddings.

Let $X = (x_1, \ldots, x_L)$ be a source code with $L$ tokens, including special tokens $[CLS]$ (class token at the first position) and $[EOS]$ (end-of-sequence token at the last position), and let $X_e \in \mathbf{R}^{L \times D}$ denote its embedding obtained from the embedding layer (i.e., $f_{plm_e}(.)$) of the used pre-trained language model. For each source code input $X$, a parameter matrix $P_X \in \mathbf{R}^{L_p \times D}$ is dynamically selected from the parameter pool $\mathcal{P}$ via the Key-Parameter Query mechanism or the Language-Aware Parameter Masking strategy. The adapted input embedding is then given by

$$X_p = \mathrm{concat}(P_X, X_e),$$

where concat denotes concatenation along the token length dimension. The resulting $X_p$ serves as the input to the PLM's multihead-attention layers $f_{plm_{mha}}(.)$. This construction enables the model to integrate **language-specific knowledge** from $P_X$ with the **shared semantic and syntactic knowledge** encoded in $X_e$, supporting more effective multilingual vulnerability detection.

In what follows, we present two elegant mechanisms for selecting $P_X$ for each input $X$, including *Parameter Selection via Key–Parameter Query* and *Language-Aware Parameter Masking*.

**Parameter Selection via Key–Parameter Query** We design a key–parameter pair-based query strategy to dynamically select the appropriate parameter for each source code input $X$. Each parameter in the pool is associated with a learnable key: $\{(k_1, P_1), (k_2, P_2), \ldots, (k_S, P_S)\}$, where each $k_i \in \mathbf{R}^{D_k}$. The set of all keys is denoted as $\mathcal{K} = \{k_i\}_{i=1}^S$. Ideally, the input itself determines which parameter to select through key–parameter matching.

This design is motivated by prior work in external memory mechanisms, i.e., VQ-VAE (van den Oord et al., 2017), where a discrete codebook is employed to retrieve task-relevant representations. Similarly, in our case, the parameter pool serves as a memory bank of language-specific knowledge, and the query mechanism enables dynamic and instance-adaptive selection of parameters.

We define a query function $q : \mathbf{R}^{L \times D} \to \mathbf{R}^{D_k}$, which maps the input to the same dimension as the keys. For simplicity, we set $D_k = D$. By default, we use the $[CLS]$ token representation obtained from the embedding layer (denoted as $f_{plm_e}(.)$) of the PLM: $q(X) = f_{plm_e}(X)_{[CLS]}$.

We denote a scoring function $\phi : \mathbf{R}^{D_k} \times \mathbf{R}^{D_k} \to \mathbf{R}$ (e.g., cosine similarity) to measure the match between the query and a key. For each input $X$, the selected parameter matrix is obtained by:

$$P_X = P_{i^*}, \quad i^* = \arg \max_{i \in [1,S]} \phi(q(X), k_i). \tag{1}$$

**Language-Aware Parameter Masking** While the default design uses instance-wise key–parameter matching, we also explore a language-aware masking strategy during training. In this approach, each language $\ell$ is associated with a fixed parameter index $i_\ell$, and the query is restricted to select only from its language-specific parameter. Formally, the selection rule for an input $X$ is:

$$P_X = P_{i^*}, \quad i^* = \arg \max_{i \in \mathcal{I}(X)} \phi(q(X), k_i), \tag{2}$$

where $\mathcal{I}(X) = \{i_\ell\}$ denotes the masked candidate determined by the language identity of $X$. This parameter assignment can be viewed as a form of supervision. Although the parameter assignment is fixed during training, the model simultaneously learns the query function $q(X)$ and the key representations $k_i$, enabling it to automatically select the appropriate parameter matrix at test time for each input $X$ using the default instance-wise key–parameter selection in Eq. (1), so the model remains language-agnostic during inference.

### 3.2.3 TRAINING OBJECTIVE FUNCTION

At each training step, after selecting the parameter matrix $P_X$ for input $X$ using the Key–Parameter Query strategy (Eq. (1) for default instance-wise selection, or Eq. (2) when Language-Aware Parameter Masking is enabled), the adapted embedding $X_p = \text{concat}(P_X, X_e)$ is fed into the multi-head attention layers $f_{plm_{mha}}(.)$ of the pre-trained language model, followed by the classifier $g_\beta(\cdot)$.

The overall training objective is to jointly optimize all model parameters through a unified loss:

$$\min_\Theta \mathcal{L}_{\text{CE}}\big(g_\beta(f_{plm_{mha}}(X_p)), Y\big) - \lambda\, \phi(q(X), k_{i^*}), \tag{3}$$

where $\Theta$ denotes all learnable parameters, including the parameter pool $\mathcal{P}$, keys $\mathcal{K}$, the pre-trained language model components $f_{plm_e}(.)$ and $f_{plm_{mha}}(.)$, and the classifier $g_\beta(.)$. $\mathcal{L}_{\text{CE}}$ is the cross-entropy loss with respect to the ground-truth label $Y$, while the second term is a surrogate loss encouraging the selected key $k_{i^*}$ to be close to the query feature $q(X)$. The scalar $\lambda$ balances the two loss terms, thereby controlling the strength of language-specific parameter specialization. Here, each input $X$ acts as a query through its representation $q(X)$, ensuring that parameter selection is directly guided by the characteristics of the source code sample.

It should be noted that the index $i^*$ of the selected parameter matrix is determined by the key–parameter rule via Eq. (1) for the default instance-wise selection, or Eq. (2) during training when language-aware masking is enabled.

### 3.2.4 A SUMMARY OF OUR MULVULN APPROACH

Algorithm 1 presents the details of our proposed MULVULN approach during both training and testing phases for multilingual vulnerability detection.

---

**Algorithm 1:** The algorithm of MULVULN for multilingual vulnerability detection.

---

**Input:** A real-world multilingual source code dataset $\mathcal{D}$ across multiple programming languages (e.g., C, C++, Java, Python, and JavaScript), consisting of $\{(X_1, Y_1), \ldots, (X_N, Y_N)\}$, where $X_i$ is a source code sample (i.e., a function) and $Y_i \in \{0, 1\}$ is its vulnerability label (0: non-vulnerable, 1: vulnerable). We denote the number of training iterations by $n_t$, the mini-batch size by $m$, and the trade-off hyper-parameter by $\lambda$. The dataset $\mathcal{D}$ is randomly partitioned into three subsets, including the training set $\mathcal{D}_{train}$ (for training the model), the validation set $\mathcal{D}_{val}$ (for model selection), and the testing set $\mathcal{D}_{test}$ (for evaluation).

1 **Training phase**

2 Initialize the keys $\{k_i\}_{i=1}^S$, the parameter pool $\{P_i\}_{i=1}^S$, and the classifier model $g_\beta(\cdot)$. Select a pre-trained language model (e.g., the encoder of CodeT5 denoted as $f_{plm}$ including $f_{plm_e}(.)$ and $f_{plm_{mha}}(.)$ as shown in Figure 1).

3 **for** $t = 1$ *to* $n_t$ **do**

4     Sample a mini-batch $\{(X_b, Y_b)\}_{b=1}^m$ from $\mathcal{D}_{train}$.

5     Obtain the embedding features $\{X_{e_b}\}_{b=1}^m$ and apply parameter selection using Key–Parameter Query (Eq. (1)) or Language-Aware Parameter Masking (Eq. (2)) to select the appropriate $P_{X_b}$ for each $X_b$, forming the adapted embeddings $\{X_{p_b}\}_{b=1}^m$.

6     Update the keys $\{k_i\}_{i=1}^S$, the parameter pool $\{P_i\}_{i=1}^S$, as well as the parameters of the pre-trained language model $f_{plm}$ and classifier $g_\beta(.)$ by minimizing the objective function (Eq. (3)) over the mini-batch using the Adam optimizer (Kingma & Ba, 2015).

7 **end**

8 **Testing phase**

9 For each input $X$ in $\mathcal{D}_{test}$, obtain its embedding $X_e$, select the parameter $P_X$ using Eq. (1), construct the adapted embedding $X_p$, and compute predictions $\hat{Y} = g_\beta(f_{plm_{mha}}(X_p))$.

**Output:** The trained model for multilingual vulnerability detection.

---

## 4 EXPERIMENTS

### 4.1 STUDIED DATASET

To evaluate our MULVULN approach and thirteen effective and state-of-the-art baselines, from deep learning to PLM-based and LLM-based approaches applied for multilingual vulnerability detection, we utilize the real-world and diverse multilingual source code REEF dataset (Wang et al., 2023a). REEF contains 4,466 CVEs with 30,987 patches across seven programming languages and provides comprehensive vulnerability information (e.g., Common Vulnerability Exposure (CVE) and Common Weakness Enumeration (CWE)) along with project metadata such as commit messages. The dataset is constructed from real-world vulnerabilities collected from the National Vulnerability Database (NVD) and Mend's CVE list (WhiteSource, 2022), from 2016 to 2023. To adapt REEF for the multilingual vulnerability detection task, we use the processed dataset from (Shu et al., 2025), which involves several preprocessing steps such as removing code comments to minimize bias and extracting vulnerable and non-vulnerable functions for each programming language, while to ensure compatibility with many PLMs relying on absolute positional encoding (typically limited to 512 tokens), functions exceeding this length are excluded. Finally, we obtained a total of 20,165 functions

with labels (i.e., vulnerable or non-vulnerable). These include 3,056 C, 1,792 C++, 427 C#, 2,905 Go, 3,235 Java, 5,468 JavaScript, and 3,282 Python functions.

We follow the same training, validation, and testing splits as in (Shu et al., 2025). Table 4 in the appendix provides detailed statistics, including the number of vulnerable and non-vulnerable functions for each programming language. In summary, the dataset contains 16,126 functions for training, 2,013 for validation, and 2,026 for testing across seven programming languages.

## 4.2 MEASURES

To measure the performance of our MULVULN approach and the baselines, we use three main metrics, commonly used in software vulnerability detection, including Recall, Precision, and F1-score (Li et al., 2018b;a; Nguyen et al., 2019; Zhou et al., 2019; Zheng et al., 2021; Nguyen et al., 2025). In the field of software vulnerability detection, F1-score (the harmonic mean of Recall and Precision) can be considered the most important metric, with Recall prioritized over Precision (Ami et al., 2024). Higher values in these metrics indicate better performances.

## 4.3 BASELINES

The baselines for our MULVULN approach consist of thirteen effective, state-of-the-art methods applied to multilingual vulnerability detection, spanning from deep learning models to large and pre-trained language models. These include TextCNN (Kim, 2014), ReGVD (Nguyen et al., 2022), CodeBERT (Feng et al., 2020), GraphCodeBERT (Guo et al., 2021), LineVul (Fu & Tantithamtha-vorn, 2022), UniXcoder (Guo et al., 2022), CodeT5 (Wang et al., 2021), CodeT5+ (Wang et al., 2023b), DeepSeek-Coder (Guo et al., 2024), Code Llama (Rozière et al., 2024), Llama 3 (Dubey et al., 2024), GPT-3.5-Turbo (OpenAI, 2022), and GPT-4o (OpenAI, 2024). We adopt different strategies depending on the model, including training from scratch (TextCNN and ReGVD), fine-tuning (GraphCodeBERT, CodeBERT, LineVul, UniXcoder, CodeT5, and CodeT5+), and zero-shot, few-shot, and instruction-based few-shot prompting (following (Shu et al., 2025)) for large language models, including DeepSeek-Coder, Code Llama, Llama 3, GPT-3.5-Turbo, and GPT-4o. To ensure fairness, all baselines and our MULVULN approach are evaluated using the same training, validation, and testing splits specified in (Shu et al., 2025), with each model trained, fine-tuned, or prompted according to its respective paradigm.

## 4.4 MODEL'S CONFIGURATIONS

For the baselines, we primarily followed the architectures and hyperparameters suggested in the corresponding papers when applying them to multilingual vulnerability detection. Furthermore, for the pre-trained language models (PLMs), we fine-tuned CodeBERT, GraphCodeBERT, the base versions of CodeT5 and CodeT5+, UniXCoder, and LineVul using the open-source checkpoints from Hugging Face (Wolf et al., 2019).

In line with (Shu et al., 2025), for experiments with closed-source LLMs, we used GPT-3.5-Turbo (model version gpt-3.5-turbo-0125) and GPT-4o (model version gpt-4o-2024-08-06) through OpenAI's API (OpenAI, 2024). For open-source LLMs, we utilized Hugging Face checkpoints (Wolf et al., 2019) for DeepSeek-Coder (6.7B parameters), Code Llama (7B parameters), and Llama 3 (8B parameters), and applied Low-Rank Adaptation (LoRA) (Hu et al., 2021) during fine-tuning to improve efficiency. For these LLMs, we employed zero-shot, few-shot, and instruction-based few-shot prompting, as in (Shu et al., 2025), and report the best results regarding F1-score.

In our MULVULN approach, Parameter Selection via Key–Parameter Query (Eq. 1) selects a single parameter matrix $P_{i^*} \in \mathbf{R}^{L_p \times D}$, with $L_p$ set to 5, a commonly used choice that balances efficiency and representational capacity. Under Language-Aware Parameter Masking, each language $\ell$ is assigned a single parameter matrix, i.e., $\mathcal{I}(X) = \{i_\ell\}$. During training, the hyperparameter $\lambda$ is tuned over $\{1 \times 10^{-1}, 3 \times 10^{-1}, 1 \times 10^{-2}, 3 \times 10^{-2}\}$, and the learning rate is fixed at $1 \times 10^{-4}$ using the Adam optimizer. For the pre-trained language model, we by default use CodeT5 (base version), one of the most effective models for vulnerability detection. All experiments were conducted on a Linux-based x86-64 machine (Precision 7865 Tower) with an AMD Ryzen Threadripper PRO 5955WX (16 cores), equipped with two RTX 6000 Ada Generation GPUs (48 GB VRAM each). The source code and data for reproducing the experiments of our MULVULN approach are publicly available in an anonymized repository at `https://anonymous.4open.science/r/mulvuln`.

## 4.5 EXPERIMENTAL RESULTS

**RQ1: How does the proposed MULVULN approach compare to thirteen effective and state-of-the-art baselines for multilingual vulnerability detection?**

We compare the performance of our MULVULN approach with thirteen effective, state-of-the-art baseline methods applied to multilingual vulnerability detection, including TextCNN, ReGVD, CodeBERT, GraphCodeBERT, LineVul, UniXcoder, CodeT5, CodeT5+, DeepSeek-Coder, Code Llama, Llama 3, GPT-3.5-Turbo, and GPT-4o, on three main popular metrics used in software vulnerability detection including Recall, Precision, and F1-score.

The experimental results in Table 1 show that our MULVULN approach, under both Parameter Selection via Key–Parameter Query and Language-Aware Parameter Masking, consistently achieves higher performance in terms of F1-score compared to the baselines. In particular, the variant with Language-Aware Parameter Masking **attains the highest F1-score of 72.20%, with improvements ranging from 1.45% to 23.59% over the baselines**. Moreover, both variants of the MULVULN approach achieve remarkably high Recall, around 97%. These results demonstrate the effectiveness of our method and its advancement in multilingual vulnerability detection.

Table 1: Performance comparison of our MULVULN approach and the baselines for multilingual vulnerability detection in terms of Recall, Precision, and F1-score. The best result for F1-score is shown in **bold**, while the second-best is shown with an underline.

| Methods | Recall | Precision | F1-score |
|---|---|---|---|
| TextCNN | 99.61% | 52.02% | 68.35% |
| ReGVD | 98.63% | 51.28% | 67.47% |
| GraphCodeBERT | 96.66% | 52.99% | 68.45% |
| CodeBERT | 100% | 51.03% | 67.57% |
| LineVul | 100% | 51.03% | 67.57% |
| UniXcoder | 89.30% | 55.18% | 68.22% |
| CodeT5 | 93.42% | 55.19% | 69.39% |
| CodeT5+ | 95.29% | 56.26% | 70.75% |
| DeepSeek-Coder | 47.89% | 49.34% | 48.61% |
| Code Llama | 91.56% | 49.50% | 64.26% |
| Llama 3 | 53.48% | 52.15% | 52.81% |
| GPT-3.5-Turbo | 61.83% | 48.88% | 54.59% |
| GPT-4o | 67.22% | 74.54% | 70.69% |
| MULVULN (w/ Eq. (1))) | 96.86% | 56.34% | 71.24% |
| MULVULN (w/ Eq. (2))) | 96.96% | 57.51% | **72.20%** |

**RQ2: How does distinct knowledge encoded via the parameter pool contribute to improving the model's performance?**

We evaluate the performance of our proposed MULVULN approach in two settings, one with the parameter pool implemented via Parameter Selection using Key–Parameter Query or Language-Aware Parameter Masking, and the other without it, using only the backbone pre-trained language model. This setup allows us to assess the impact of distinct knowledge encoded via the parameter pool on multilingual vulnerability detection in terms of Recall, Precision, and F1-score. In this ablation study, we use the encoder of CodeT5 (base version) or CodeT5+ (base version), two of the most effective PLMs for software vulnerability detection, as the backbone of our MULVULN approach.

The results in Table 2 clearly demonstrate the impact of the parameter pool on model performance. The encoded distinct knowledge through the parameter pool significantly improves performance in terms of Recall and F1-score, highlighting the effectiveness of our approach. For instance, compared to CodeT5, MULVULN with Language-Aware Parameter Masking achieves improvements of **3.54% and 2.81% in Recall and F1-score**, respectively. Similarly, compared to CodeT5+, MULVULN using either Parameter Selection via Key–Parameter Query or Language-Aware Parameter Masking consistently achieves gains in both Recall and F1-score.

Table 2: Performance comparison of our MULVULN approach with the parameter pool, encoding distinct knowledge, and without it, using only the backbone pre-trained models CodeT5 or CodeT5+, for multilingual vulnerability detection in terms of Recall, Precision, and F1-score. The best results are shown in **bold**.

| Methods | Recall | Precision | F1-score |
|---|---|---|---|
| CodeT5 | 93.42% | 55.19% | 69.39% |
| MULVULN-CodeT5 (w/ Eq. (1)) | 96.86% | 56.34% | 71.24% |
| MULVULN-CodeT5 (w/ Eq. (2))) | **96.96%** | **57.51%** | **72.20%** |
| CodeT5+ | 95.29% | 56.26% | 70.75% |
| MULVULN-CodeT5+ (w/ Eq. (1))) | 96.96% | **56.36%** | **71.28%** |
| MULVULN-CodeT5+ (w/ Eq. (2))) | **99.31%** | 55.48% | 71.19% |

**RQ3: How does MULVULN perform on the top-10 critical CWEs?**

We evaluate the performance of our proposed MULVULN approach on the top-10 CWEs, following the latest 2024 MITRE Top 25 scoring[1], which considers prevalence, exploitability, impact, and current industry perception. Our analysis focuses on Recall and F1-score, the two most important and prioritized metrics in software vulnerability detection. The experimental results in Table 3 further demonstrate the effectiveness and reliability of MULVULN. For the top-10 CWEs, the testing subset contains 657 samples, including 346 labeled as vulnerable. On this subset, MULVULN achieves an average Recall of 96.27% and an F1-score of 71.82%.

Table 3: Performance of our MULVULN approach on the top-10 CWEs in terms of Recall and F1-score, including the number of vulnerable and total samples (both vulnerable and non-vulnerable).

| CWEs | Recall | F1-score | Vul Samples | Total Samples |
|---|---|---|---|---|
| CWE-79 (Cross-Site Scripting) | 94.85% | 73.31% | 97 | 198 |
| CWE-787 (Out-of-Bounds Write) | 100% | 70.97% | 22 | 41 |
| CWE-89 (SQL Injection) | 90.91% | 66.67% | 22 | 46 |
| CWE-78 (OS Command Injection) | 100% | 52.63% | 5 | 15 |
| CWE-416 (Use After Free) | 94.74% | 70.59% | 19 | 34 |
| CWE-20 (Improper Input Validation) | 96.15% | 73.53% | 52 | 92 |
| CWE-125 (Out-of-Bounds Read) | 100% | 77.11% | 32 | 51 |
| CWE-22 (Path Traversal) | 97.50% | 70.27% | 40 | 76 |
| CWE-352 (Cross-Site Request Forgery) | 95.24% | 80.81% | 42 | 80 |
| CWE-94 (Code Injection) | 93.33% | 82.35% | 15 | 24 |
| **Average** | 96.27% | 71.82% | 346 | 657 |

## 5 CONCLUSION

In this paper, we introduce MULVULN, an innovative deep learning-based approach for multilingual vulnerability detection. Our MULVULN framework is designed to enhance pre-trained language models, which generalize across languages and capture semantic and syntactic relationships crucial for vulnerability detection, by introducing a parameter pool to model language-specific features. Together, these parts enable the model to generalize across diverse programming languages while adapting to their unique characteristics, providing a more robust and effective solution for multilingual vulnerability detection. Extensive experiments on the real-world and diverse source code REEF dataset demonstrate the effectiveness of MULVULN, showing consistent and significant improvements over thirteen strong state-of-the-art baselines. In particular, our approach achieves the best performance on F1-score and strong performance on Recall, the two key metrics prioritized in software vulnerability detection.

---

[1]https://cwe.mitre.org/top25/

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

# A APPENDIX

## A.1 DATASET STATISTICS

To provide a clearer understanding of the real-world and diverse multilingual REEF dataset used in our experiments, Table 4 presents its statistical summary. It reports the number of functions in the training, validation, and testing sets, as well as the distribution of vulnerable and non-vulnerable samples across seven programming languages. For clarity, the languages are sorted in ascending order based on the total number of samples.

Table 4: Statistical summary of the REEF dataset.

| Languages | Training | Validation | Test | Vul | Non-Vul | Total |
|---|---|---|---|---|---|---|
| C# | 341 | 42 | 44 | 212 | 215 | 427 |
| C++ | 1,432 | 179 | 181 | 911 | 881 | 1,792 |
| Go | 2,323 | 290 | 292 | 1,462 | 1,443 | 2,905 |
| C | 2,444 | 305 | 307 | 1,541 | 1,515 | 3,056 |
| Java | 2,587 | 323 | 325 | 1,622 | 1,613 | 3,235 |
| Python | 2,625 | 328 | 329 | 1,642 | 1,640 | 3,282 |
| JavaScript | 4,374 | 546 | 548 | 2,743 | 2,725 | 5,468 |
| **Total** | 16,126 | 2,013 | 2,026 | 10,133 | 10,032 | 20,165 |

## A.2 ADDITIONAL EXPERIMENTS

### A.2.1 PERFORMANCE OF MULVULN BY PROGRAMMING LANGUAGE

In this section, we evaluate the performance of our MULVULN approach using the Language-Aware Parameter Masking strategy (Eq. (2)). As shown in Table 1, it achieves the best F1-score across different programming languages. The results, summarized in Table 5, show that MULVULN achieves the highest Precision and F1-score, 65.68% and 78.24%, respectively, on *JavaScript*. Moreover, for *C*, it attains the highest Recall of 100%, demonstrating its effectiveness across diverse languages.

Table 5: Performance of our proposed MULVULN approach on Recall, Precision, and F1-score metrics by programming language.

| Languages | Recall | Precision | F1-score |
|---|---|---|---|
| C# | 95.45% | 60.00% | 73.68% |
| C++ | 98.91% | 52.91% | 68.94% |
| Go | 98.64% | 58.70% | 73.60% |
| C | **100%** | 53.08% | 69.35% |
| Java | 96.93% | 54.67% | 69.91% |
| Python | 92.12% | 54.68% | 68.62% |
| JavaScript | 96.73% | **65.68%** | **78.24%** |

### A.2.2 VISUALIZING LANGUAGE-SPECIFIC PARAMETERS AND QUERY DISTRIBUTIONS

We analyze how the parameter pool interacts with input queries and demonstrate the effectiveness of the Parameter Selection via Key–Parameter Query mechanism (Eq. (1)) and the Language-Aware Parameter Masking strategy (Eq. (2)) in aligning queries with their corresponding language-specific parameters on test samples after training. The selected parameters are then combined with the input embeddings in the PLM's multihead-attention layers, allowing the model to integrate shared knowledge captured by the PLM with enhanced language-specific information, supporting more effective multilingual vulnerability detection.

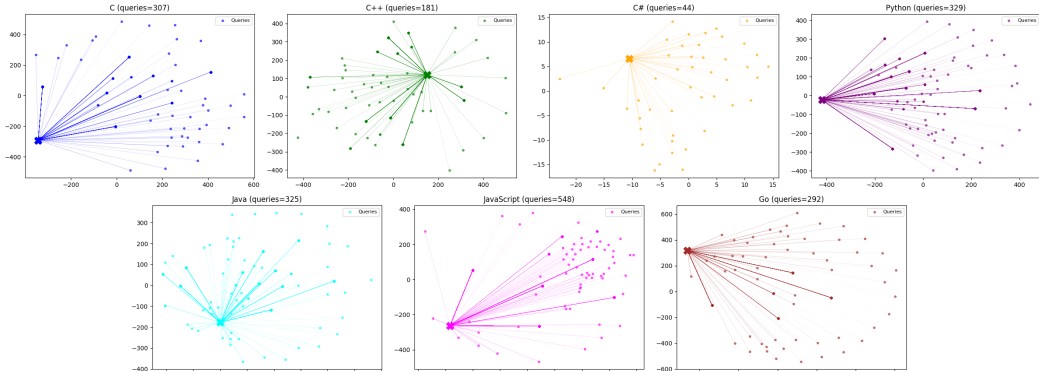

Figure 2: Visualization of the parameter pool and queries using t-SNE under the Parameter Selection via Key–Parameter Query mechanism (Eq. (1)). Each subplot corresponds to a different programming language. The × marker represents the parameter, and scatter points are queries from test samples of each language. Arrows indicate instance-wise key–parameter associations. Queries radiate outward from their parameter, forming "peacock tail" patterns that reflect sample-level diversity while maintaining a stable language-specific reference.

Figures 2 and 3 show the parameter pool and queries for all test samples of each programming language. In Figure 2, under the Parameter Selection via Key–Parameter Query mechanism (Eq. (1)), each subplot shows that the parameter (× marker) acts as an anchor, with queries radiating outward to form "peacock tail" patterns that reflect sample-level diversity while maintaining a stable language-specific reference. In Figure 3, under the Language-Aware Parameter Masking strategy (Eq. (2)), queries remain anchored to their corresponding parameter and are generally oriented toward it while preserving distinctions between individual samples. However, for C#, the parameter is positioned farther from its queries, probably due to the limited number of training samples (around 341 versus thousands for other languages), illustrating weaker parameter–query alignment despite cosine similarity-based selection. Notably, this issue does not occur in Eq. (1), which can be attributed to the ability of queries to select from the entire parameter pool, allowing dynamic adjustment even for underrepresented languages. In contrast, languages with more training samples form tighter, more concentrated clusters.

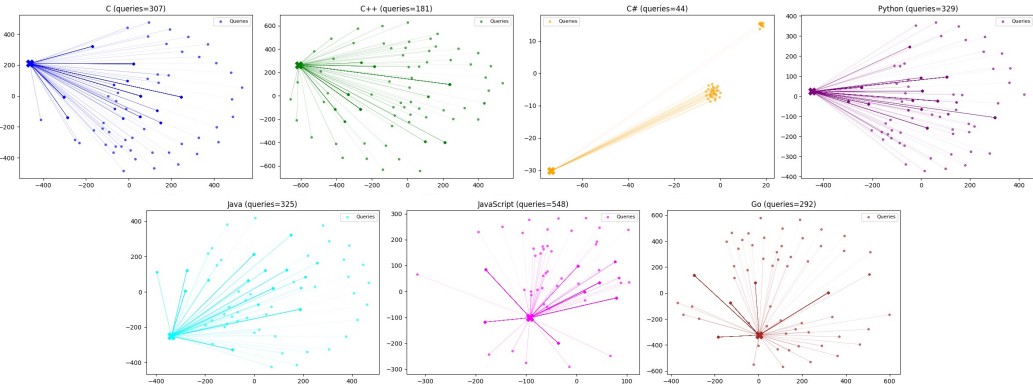

Figure 3: Visualization of the parameter pool and queries using t-SNE under the Language-Aware Parameter Masking strategy (Eq. (2)). Each subplot corresponds to a different programming language. Queries remain anchored to their parameter, generally oriented toward it while preserving distinctions between individual samples. For C#, the parameter is positioned farther from its queries, probably due to limited training samples (around 341 versus thousands for other languages), illustrating weaker parameter–query alignment despite cosine similarity-based selection.

Overall, these visualizations indicate that the parameter pool provides a consistent reference for language-specific knowledge, while the distribution of queries shows that the PLM preserves shared knowledge across languages without collapsing the diversity of individual query embeddings.

A.3   THREATS TO VALIDITY

**Construct Validity**   A key threat to construct validity lies in whether our assessments accurately capture the ability of the methods to perform multilingual vulnerability detection. The primary goal of our MULVULN approach is to address this problem in the real-world, challenging, and diverse multilingual source code dataset. To evaluate the performance of MULVULN and the baselines, we use three main measures, commonly used in software vulnerability detection, including Recall, Precision, and F1-score (Li et al., 2018b;a; Nguyen et al., 2019; Zhou et al., 2019; Zheng et al., 2021; Nguyen et al., 2025). In the field of software vulnerability detection, F1-score can be considered the most important metric, with Recall typically prioritized over Precision (Ami et al., 2024).

**Internal Validity**   Internal validity threats mainly relate to the choice of hyperparameter settings (e.g., optimizer, learning rate, and the number of layers in deep neural networks). Finding optimal hyperparameter configurations is often expensive due to the large number of trainable parameters. In training MULVULN, we generally adopt widely used values, such as the Adam optimizer with a learning rate of $1 \times 10^{-4}$. For Parameter Selection via Key–Parameter Query, a single parameter matrix $P_{i*} \in \mathbf{R}^{L_p \times D}$ is selected for each input (Eq. (1)) with $L_p$ set to 5, a commonly used choice that balances efficiency and representational capacity, and $D$ corresponding to the embedding size of the pre-trained model (i.e., CodeT5 (base version)). For Language-Aware Parameter Masking, each language $\ell$ is associated with a fixed parameter matrix index $i_\ell$ (Eq. (2)). All hyperparameter settings are reported in our released reproducible source code to support future replication studies.

**External Validity**   External validity threats concern whether MULVULN can generalize effectively to real-world and diverse multilingual source code vulnerabilities. We mitigate this by conducting rigorous experiments on the multilingual REEF dataset, which contains 4,466 CVEs with 30,987 patches across seven programming languages. REEF provides comprehensive vulnerability information (e.g., Common Vulnerability Exposure (CVE) and Common Weakness Enumeration (CWE), and Common Vulnerability Scoring System (CVSS)) along with project metadata such as commit messages, and is constructed from real-world vulnerabilities collected from the National Vulnerability Database (NVD) and Mend's CVE list (WhiteSource, 2022), covering the years 2016–2023. The experimental results demonstrate that MULVULN outperforms the baselines by a wide margin, particularly in F1-score.

A.4   THE FLEXIBILITY OF OUR PROPOSED MULVULN APPROACH

Our MULVULN approach is flexible and can be applied to both transformer-based encoder–decoder and encoder-only pre-trained LLMs. In this work, we leverage only the encoder component of pre-trained language models, since the detection task primarily requires understanding and representing source code, which is best captured by the encoder. In default, our proposed MULVULN approach uses CodeT5, a widely adopted pre-trained model for software vulnerability detection. In the ablation study for RQ2 (presented in Section 4.5), we also apply MULVULN with CodeT5+, demonstrating that our framework can readily adapt to different pre-trained language models.

A.5   ABLATION STUDIES

A.5.1   IMPACT OF PARAMETER LENGTH ($L_p$) ON MULVULN PERFORMANCE

We conducted an ablation study to evaluate the effect of varying the parameter length $L_p \in \{1, 3, 5, 7, 9\}$ on the performance of our MULVULN approach under both Parameter Selection via Key–Parameter Query (Eq. (1)) and Language-Aware Parameter Masking (Eq. (2)). The results are summarized in Table 6. Performance evaluation and conclusions are based on F1-score, the harmonic mean of Precision and Recall, which balances the two metrics.

To better interpret the impact of parameter length, we categorize $L_p$ into small, intermediate, and large values and discuss their effects on model performance:

- **Small values ($L_p = 1, 3$):** These settings seem to limit representational capacity, resulting in lower F1-scores despite high Recall. Notably, $L_p = 3$ achieves the highest Precision for Eq. (1), but its F1-score remains lower than that of $L_p = 5$.

Table 6: Ablation study on parameter length ($L_p$) for our proposed MULVULN approach under both Key–Parameter Query (Eq. (1)) and Language-Aware Parameter Masking (Eq. (2)). The best results in each mechanism are highlighted in **bold**.

| Methods | $L_p$ | Recall | Precision | F1-score |
|---|---|---|---|---|
| MULVULN w/ Eq. (1) | 1 | 90.97% | 56.15% | 69.44% |
| MULVULN w/ Eq. (1) | 3 | 86.95% | **57.91%** | 69.52% |
| MULVULN w/ Eq. (1) | 5 | **96.86%** | 56.34% | **71.24%** |
| MULVULN w/ Eq. (1) | 7 | 95.00% | 56.94% | 71.20% |
| MULVULN w/ Eq. (1) | 9 | 94.31% | 56.13% | 70.38% |
| MULVULN w/ Eq. (2) | 1 | 91.95% | 56.04% | 69.64% |
| MULVULN w/ Eq. (2) | 3 | 96.07% | 56.07% | 70.81% |
| MULVULN w/ Eq. (2) | 5 | **96.96%** | **57.51%** | **72.20%** |
| MULVULN w/ Eq. (2) | 7 | 95.29% | 57.05% | 71.37% |
| MULVULN w/ Eq. (2) | 9 | 89.01% | 56.65% | 69.24% |

- **Intermediate value ($L_p = 5$):** This configuration achieves the highest F1-score across both mechanisms (i.e., Key–Parameter Query (Eq. (1)) and Language-Aware Parameter Masking (Eq. (2))), striking a balance between sufficient parameter-specific representations and avoiding redundancy.

- **Large values ($L_p = 7$ or $9$):** For Eq. (1), Precision increases at $L_p = 7$ compared to $L_p = 5$ but decreases at $L_p = 9$. For Eq. (2), Precision shows a decline at $L_p = 7$ and $L_p = 9$. Recall and F1-score decrease for both mechanisms at these large $L_p$ values. These trends may result from over-parameterization, where additional parameters introduce redundancy rather than meaningful information, reducing overall performance.

**Overall finding:** Based on F1-score, $L_p = 5$ provides the most effective trade-off between representational capacity, efficiency, and generalization.

### A.5.2 PARAMETER SELECTION STRATEGIES

Our parameter pool is designed to encode the distinct knowledge of each programming language, with each language primarily using its own dedicated parameter matrix. By default, MULVULN selects a single parameter matrix $P_{i*}$ for each input $X$ via Key–Parameter Query (Eq. (1)), ensuring dedicated representations while leveraging shared knowledge from the pre-trained model. To better understand the effects of flexible selection, we conduct multi-parameter ablation studies.

**Multi-Parameter Selection via Key–Parameter Query** We explore selecting the top-$K$ matching keys ($K > 1$) for a single input, allowing $X$ to leverage both its distinct parameter and additional shared matrices with other inputs, and examine how this affects the performance of our MULVULN approach.

**Multi-Parameter Extension in Language-Aware Parameter Masking** In this setting, we consider a multi-parameter extension for Language-Aware Parameter Masking (Eq. (2)). Specifically, each language $\ell$ can be associated with multiple parameter matrices instead of just one. Inputs from the same language select among these matrices, allowing us to study the effect of increasing language-specific capacity while preserving language-specific distinctions.

**Impact of Multi-Parameter Selection on Performance** Experimental results in Table 7 show that the single-parameter setting provides the best trade-off between specialization and generalization. For Key–Parameter Query, using multiple parameter matrices improves Recall but reduces Precision, leading to an overall drop in F1-score compared to the default single-parameter setup. For Language-Aware Parameter Masking, the single-parameter setting achieves the highest F1-score, while increasing the number of matrices consistently degrades performance.

Table 7: Ablation study of our proposed MULVULN approach on parameter selection strategies. Eq. (1) corresponds to Key–Parameter Query, and Eq. (2) corresponds to Language-Aware Parameter Masking. Here, *pm* and *pms* denote parameter matrix and parameter matrices, respectively.

| Methods | Recall | Precision | F1-score |
|---|---|---|---|
| MULVULN (1 pm) w/ Eq. (1) | 96.86% | **56.34%** | **71.24%** |
| MULVULN (2 pms) w/ Eq. (1) | 98.43% | 54.69% | 70.31% |
| MULVULN (3 pms) w/ Eq. (1) | **98.53%** | 54.54% | 70.21% |
| MULVULN (1 pm) w/ Eq. (2) | **96.96%** | 57.51% | 72.20% |
| MULVULN (2 pms) w/ Eq. (2) | 95.29% | 56.42% | 70.88% |
| MULVULN (3 pms) w/ Eq. (2) | 93.13% | 56.52% | 70.35% |

These outcomes can be attributed to several factors. *Multi-Parameter Extension in Language-Aware Parameter Masking* may blur language-specific distinctions and increase model capacity, potentially leading to overfitting, slower convergence, and more challenging optimization. *Multi-Parameter Selection via Key–Parameter Query* can increase the risk of suboptimal combinations due to noisy selection when multiple top-$K$ keys are chosen for a single input, some parameters may not perfectly match, introducing conflicting signals that reduce the effectiveness of distinct knowledge. Furthermore, since the pre-trained model already captures shared cross-language knowledge, additional instance-wise parameter sharing can be redundant and may dilute useful signals.

Overall, these findings exhibit the effectiveness of the single-parameter matrix selection configuration, as it preserves language-specific knowledge while leveraging shared pre-trained representations, providing a clean adapter mechanism without unnecessary complexity.

**Future Directions** While the experimental results favor the single-parameter matrix selection setting, future research could explore adaptive strategies that combine the benefits of single-parameter and multi-parameter approaches. For instance, dynamically adjusting the number of parameters per input based on language complexity, data availability, or task difficulty may mitigate the limitations of fixed multi-parameter selection. Another promising direction is lightweight regularization or gating mechanisms that selectively control parameter sharing, achieving a better balance between specialization and generalization.

