# OpenReview forum: "MulVuln: Enhancing Pre-trained LLMs with Shared and Language-Specific Knowledge for Multilingual Vulnerability Detection"
_ICLR.cc/2026/Conference — Submitted to ICLR 2026_

### Official Review · Reviewer_YvFQ · 2025-10-19

**Soundness:** 2
**Presentation:** 3
**Contribution:** 2
**Rating:** 4
**Confidence:** 4

**Summary:**

The paper proposes a method for enhancing software vulnerability detection by generalizing across multiple programming languages. The proposed solution utilizes CodeT5, a pretrained language model (PLM) as a backbone that includes general parametric knowledge of multiple languages. The input to the PLM is then augmented by several extra tokens designed to signal which programming language is being used. The embeddings for those extra tokens are learned during the training of the model, and the model further incentivized to select the embeddings associated with the correct language through two proposed approaches: key parameter query, and parameter masking.

**Strengths:**

- The proposed method seems like a lightweight add-on to pretrained language models that slightly improves their performance on vulnerability detection, and may be relevant for other software-related tasks as well.
- The “parameter selection via key-parameter query” method seems interesting as it bears resemblance to the attention mechanism in transformer models, despite the fact that it showed less impressive results than the “language-aware parameter masking”

**Weaknesses:**

- The related work section did not cite any of the existing works targeting multi-lingual software vulnerability detection (e.g., [A]. [B], [C]. These paper should have been discussed in the related work, and even used as baselines for comparison.

- The authors mentioned polyglot applications (i.e., projects including multiple languages) to motivate their proposed solution, but this type of software was not present in the considered dataset and was thus never evaluated in the experiments.

- The external validity of the work is questionable. Although the proposed detector was trained on the training set of the REEF dataset, and was evaluated on the test set of the same dataset, this might not be enough. Other datasets (from the plethora of available datasets on vulnerability detection) should have been considered to prove the efficacy of the proposed method in various settings.

[A] Zhang, Boyu, Triet HM Le, and M. Ali Babar. "MVD: A Multi-Lingual Software Vulnerability Detection Framework." *arXiv preprint arXiv:2412.06166* (2024).

[B] Zhang, Ting, et al. "Benchmarking Large Language Models for Multi-Language Software Vulnerability Detection." *arXiv preprint arXiv:2503.01449* (2025).

[C] Yu, Junji, et al. "A Preliminary Study of Large Language Models for Multilingual Vulnerability Detection." *Proceedings of the 34th ACM SIGSOFT International Symposium on Software Testing and Analysis*. 2025.

**Questions:**

- Why did the authors not cite related works on multi-lingual vulnerability detection in their related work? and why were performance comparisons not conducted with those recent works?

- In section 3.1 you mention that you consider function-level binary classification of vulnerabilities. However this exact problem has come under strong criticism from recent work [A], namely it fails to incorporate contextual information which are crucial to deciding whether software is vulnerable. How would you defend this choice?

- For multi-lingual projects, could a weighted sum of language-specific parameters $P_X$ be used instead of the argmax in equations (1) ad (2)?

- For pretrained language models, have you experimented with chain-of-thought prompting? You could  elicit the model to first output the used programming language, which would naturally condition the upcoming generated tokens deciding whether the code is vulnerable. It would be interesting to compare this to the learnable conditioning proposed by your solution.


[A] Risse, Niklas, Jing Liu, and Marcel Böhme. "Top score on the wrong exam: On benchmarking in machine learning for vulnerability detection." *Proceedings of the ACM on Software Engineering* 2.ISSTA (2025): 388-410.

---

> ### Author Response · Authors · 2025-11-21
> **Authors’ responses to Reviewer YvFQ (Part 1)**
>
> **We thank the reviewer for reviewing our paper and providing comments and questions for further enhancements and clarifications. Below, we provide responses to all the comments and questions.**
>
> **Q.** *Why did the authors not cite related works on multi-lingual vulnerability detection in their related work? and why were performance comparisons not conducted with those recent works?*
>
> **Response.** In our paper, [C]’s extension, Shu et al., 2025, is one of the main related works we cite in both the related work and experiment sections. Shu et al., 2025, comes from the same research group as [C] and provides a newer and more comprehensive update and experiments, which is why we primarily reference it in our experiments. The dataset used in Shu et al., 2025, and adopted in our study covers a larger set of programming languages (compared to [B]) and includes well-defined training, validation, and test splits, enabling a thorough evaluation of multilingual vulnerability detection methods.
>
> Papers [A] (which introduces a method for multilingual vulnerability detection) and [B] (which introduces a dataset for multilingual vulnerability detection and evaluates some LLM-based methods) are currently available as preprints on arXiv, and we could not find evidence that they have undergone peer review. While relevant, *our study focuses on established datasets and peer-reviewed methods to ensure fair and rigorous comparison*. Furthermore, many of the LLM-based methods evaluated in [B] are also included in our experiments.
>
> The chosen dataset, together with the 13 effective and state-of-the-art methods from prior published work that we use in our paper, provides a comprehensive and representative evaluation of current approaches to multilingual vulnerability detection. We believe this choice ensures the reliability and comparability of our results while highlighting the contributions of our proposed MulVuln approach.
>
> **Q.** *In section 3.1 you mention that you consider function-level binary classification of vulnerabilities. However this exact problem has come under strong criticism from recent work (Risse et al., 2023), namely it fails to incorporate contextual information which are crucial to deciding whether software is vulnerable. How would you defend this choice?*
>
> **Response.** We acknowledge the concern raised by (Risse et al., 2023) regarding function-level vulnerability classification and the potential need for broader contextual information. However, in many real-world cases, individual functions can be vulnerable independently, making function-level analysis both meaningful and practical. Most existing vulnerability detection methods (e.g., the well-known and state-of-the-art baselines used in our study) still primarily focus on this level.
>
> Function-level detection is suitable for multilingual vulnerability detection, as it allows the model to focus on fine-grained semantic and syntactic patterns across diverse programming languages, which is essential for learning both shared and language-specific representations.
>
> Using function-level evaluation ensures consistency with prior work and the baselines included in our study, enabling fair and direct comparisons. While our experiments focus on this level, the MulVuln architecture is general and can be extended to incorporate broader program-level or system-level context, representing a promising direction for future work.
>
> **Q.** *For multi-lingual projects, could a weighted sum of language-specific parameters be used instead of the argmax in equations (1) and (2)?*
>
> **Response.** In our study, the parameter pool is specifically designed to encode language-specific knowledge, and its size is intentionally set to match the number of programming languages in the dataset. This aligns with the pool’s purpose of representing per-language knowledge. Accordingly, we select a single parameter matrix for each input X via *the argmax in Eqs. (1) and (2)*, which ensures that the most relevant language-specific knowledge is applied.
>
> While it is theoretically possible to use a weighted sum or concatenation of multiple parameter matrices (allowing a soft alternative to the discrete argmax selection or an increase in language-specific capacity) for a single input X, our ablation study (Appendix A.5.2) shows that selecting a single parameter matrix per input X achieves the best performance on the used dataset. In our empirical evaluation, using multiple parameters did not show clear benefits and can introduce additional complexity, which may affect how language-specific features are modeled.
>
> Overall, these findings demonstrate that the single-parameter selection strategy effectively preserves language-specific knowledge while leveraging shared cross-language knowledge encoded by a pre-trained language model’s encoder (e.g., CodeT5’s encoder), providing a clean and efficient mechanism for multilingual vulnerability detection.

---

> ### Author Response · Authors · 2025-11-21
> **Authors’ responses to Reviewer YvFQ (Part 2)**
>
> **Q.** *For pretrained language models, have you experimented with chain-of-thought prompting? You could elicit the model to first output the used programming language, which would naturally condition the upcoming generated tokens deciding whether the code is vulnerable. It would be interesting to compare this to the learnable conditioning proposed by your solution.*
>
> **Response.** Thank you for the suggestion. Chain-of-thought prompting to first identify the language and then predict vulnerabilities could be an interesting idea. However, its effective application generally requires datasets specifically designed to provide reliable intermediate reasoning, especially for the vulnerability prediction phase. Most datasets in software vulnerability detection, including the multilingual REEF dataset used in our experiments with standard splits (Shu et al., 2025), are not structured and do not provide annotations for intermediate reasoning steps, which, to our knowledge, is the most challenging part of applying chain-of-thought prompting for vulnerability prediction.
>
> In our paper, we follow standard and state-of-the-art training procedures for deep learning, pre-trained, and large language models, including full supervised training, supervised fine-tuning, and various prompting strategies (zero-shot, few-shot, instruction fine-tuning), ensuring a fair and rigorous evaluation.
>
> Importantly, our MulVuln approach, when using the Parameter Selection via Key–Parameter Query mechanism in the training phase, is language-agnostic, as it does not require the programming language to be known beforehand (it can operate on input code without needing to know which programming language it is written in). Moreover, our approach does not require language identification at inference time. Specifically, the model leverages a parameter pool to encode language-specific knowledge while simultaneously capturing shared cross-language semantic and syntactic representations. This enables the model to automatically adapt to the characteristics of input code across different programming languages, without an explicit language prediction step.
>
> **Q.** *The authors mentioned polyglot applications (i.e., projects including multiple languages) to motivate their proposed solution, but this type of software was not present in the considered dataset and was thus never evaluated in the experiments.*
>
> **Response.** Given the availability of publicly accessible benchmark datasets, we follow the standard evaluation protocol in the field by conducting experiments on a real-world, diverse multilingual source code dataset, REEF, to mimic scenarios where code written in different programming languages coexists in the same project. This approach aligns with prior multilingual vulnerability detection work (e.g., Shu et al., 2025) and effectively reflects general real-world scenarios. Our experiments demonstrate the effectiveness of both the proposed MulVuln method and the baselines in addressing the multilingual vulnerability detection problem.
>
> **Q.** *Although the proposed detector was trained on the training set of the REEF dataset, and was evaluated on the test set of the same dataset, this might not be enough.*
>
> **Response.** Regarding generalization and realistic evaluation, the REEF dataset contains 4,466 CVEs with 30,987 patches across seven programming languages, collected from real-world vulnerabilities in the National Vulnerability Database (NVD) and Mend’s CVE list, spanning 2016–2023. This dataset captures substantial diversity in programming languages, code patterns, and temporal coverage, providing a rigorous test of the model’s ability to generalize to previously unseen code samples.
>
> In the pre-defined splits of the training, validation, and test sets (as used in our experiments and in prior work, e.g., Shu et al., 2025), the test set contains source code samples that do not appear in the training set. Together with the coverage of multiple programming languages, code structures, and CVE types, this setup ensures a realistic and diverse evaluation of the model’s performance, demonstrating the effectiveness of our proposed approach in practical multilingual vulnerability detection.

---

### Official Review · Reviewer_SqYM · 2025-10-25

**Soundness:** 3
**Presentation:** 3
**Contribution:** 2
**Rating:** 4
**Confidence:** 3

**Summary:**

This paper addresses the problem of multilingual software vulnerability detection by proposing MulVuln, a framework that augments pre-trained language models with a parameter pool specifically designed to capture both shared (cross-lingual) and language-specific knowledge in source code. The approach combines the strengths of PLMs for general semantic representation with dynamic selection of language-tailored parameters via either key-parameter querying or language-aware masking. The method is rigorously evaluated against thirteen baselines on the challenging REEF dataset, demonstrating superior F1 and recall, as well as strong performance on top vulnerability types (CWEs) and across seven diverse programming languages.

**Strengths:**

- The proposed method consistently outperforms a wide range of strong baselines and modern LLMs.
- The paper provides explicit interpretability through visualizations that offer valuable insights into the inner workings of the model.
- The research ensures reproducibility and offers practical value by providing sufficient implementation details and using a realistic dataset.

**Weaknesses:**

- Although the performance of multilingual code vulnerability detection is commendable, some related work on vulnerability detection based on LLMs  (beyond merely utilizing LLMs themselves) has not been cited or thoroughly analyzed. Due to the lack of a clear comparison with LLM-based vulnerability detection work, its originality is limited.
- There is limited analysis of the computational overhead. The design introduces language-specific parameter matrices on top of parameter-rich PLMs, which may incur additional computational costs.
- Although recall and F1 are highlighted (arguably reasonable in vulnerability detection due to prioritizing recall), MulVuln’s precision often increases less than recall. In Table 1, some baselines achieve comparable or even higher precision. Given the application (where false positives may incur costs), more discussion or tuning toward precision-oriented use cases is warranted.

**Questions:**

- The main benefit is claimed around the dynamic query mechanism for associating code (possibly ambiguous or mixed-language) with the right parameter. How does the model behave for code samples with mixed or embedded scripting languages, or where the language cannot be reliably determined upfront? Are there empirical results for such edge cases?
- Results currently lack confidence intervals/statistical significance analysis. Are the observed improvements in F1-score over the best PLM/LLM baselines robust to different seeds or test splits? Could the authors report mean and variance over multiple runs?

---

> ### Author Response · Authors · 2025-11-21
> **Authors’ responses to Reviewer SqYM (Part 1)**
>
> **We thank the reviewer for reviewing our paper and providing comments and questions for further enhancements and clarifications. Below, we provide responses to all the comments and questions.**
>
> **Q.** *Although the performance of multilingual code vulnerability detection is commendable, some related work on vulnerability detection based on LLMs (beyond merely utilizing LLMs themselves) has not been cited or thoroughly analyzed. Due to the lack of a clear comparison with LLM-based vulnerability detection work, its originality is limited.*
>
> **Response.** In the experiments, we compare our MulVuln approach with a variety of methods that have demonstrated strong performance in software vulnerability detection, as well as the ability to handle multilingual vulnerability detection.
>
> In particular, we compare our method against 13 effective and state-of-the-art baselines, including well-known pre-trained and large language models such as CodeBERT, GraphCodeBERT, LineVul, UniXcoder, CodeT5, CodeT5+, DeepSeek-Coder, Code Llama, Llama 3, GPT-3.5-Turbo, and GPT-4o.
>
> We would like to highlight that the LLM-based methods included in our experiments were adapted for the task using diverse, effective approaches, as used in prior work. For example, CodeT5 and CodeT5+ were fine-tuned using their encoder components followed by a multi-layer feedforward classifier head, while models such as DeepSeek-Coder, Code Llama, Llama 3, GPT-3.5-Turbo, and GPT-4o were instruction fine-tuned in addition to prompt engineering. These adaptations enable the models to perform vulnerability detection effectively while leveraging their inherent ability to handle the multilingual setting.
>
> This set of baselines (i.e., 13 strong, widely adopted, and recent methods) provides a broad and representative evaluation of current approaches applied in multilingual vulnerability detection.
>
> **Q.** *There is limited analysis of the computational overhead. The design introduces language-specific parameter matrices on top of parameter-rich PLMs, which may incur additional computational costs.*
>
> **Response.** In our experiments, to demonstrate the effectiveness of our proposed approach while maintaining computational efficiency, we used the base versions (220M parameters) of CodeT5 and CodeT5+. Although MulVuln introduces language-specific parameter matrices, the additional overhead is modest, approximately $\(27\text{k}\)$ parameters, given a parameter pool size $\(\mathcal{P} = 7\)$, parameter length $\(L\_p = 5\)$, and embedding dimension $\(D = 768\)$. This overhead is negligible compared to the overall model size. Moreover, when compared to large language models with billions of parameters, our approach remains significantly more efficient in terms of both memory usage and inference time.
>
> **Q.** *Although recall and F1 are highlighted (arguably reasonable in vulnerability detection due to prioritizing recall), MulVuln’s precision often increases less than recall. In Table 1, some baselines achieve comparable or even higher precision. Given the application (where false positives may incur costs), more discussion or tuning toward precision-oriented use cases is warranted.*
>
> **Response.** Our MulVuln approach achieves a much higher Recall and F1-score (the two key and prioritized metrics in software vulnerability detection), while its Precision is lower than some baselines. This reflects the inherent trade-off between Precision and Recall: A model with higher Precision may be more conservative, making fewer positive predictions and thereby reducing false positives, but often at the expense of Recall.
>
> The lower Precision to some baselines, along with a much higher Recall compared to all the baselines, indicates that our MulVuln approach classifies more instances as positive (i.e., vulnerabilities), potentially increasing false positives but also capturing more true positives. This trade-off reflects the model’s emphasis on Recall, a priority in security contexts, where missing vulnerabilities (false negatives) can lead to severe consequences.
>
> The challenge of achieving a good balance between Precision and Recall in vulnerability detection is amplified on the challenging real-world dataset, such as the REEF dataset used in our experiment, where vulnerability patterns vary widely across scenarios, programming languages, and CVE types, and where there is diversity between the training and testing sets.
>
> In future work, we aim to improve the model’s generalization to achieve higher performance in Recall, F1-score, and Precision. One potential direction is to create additional diverse augmented data for the training data sets, which can help the model better distinguish between true vulnerabilities and non-vulnerabilities. We also aim to explore fine-tuning strategies to improve the balance between Precision and Recall, thereby enhancing the model’s accuracy and reliability in real-world security applications.

---

> ### Author Response · Authors · 2025-11-21
> **Authors’ responses to Reviewer SqYM (Part 2)**
>
> **Q.** *The main benefit is claimed around the dynamic query mechanism for associating code with the right parameter. How does the model behave for code samples with mixed or embedded scripting languages, or where the language cannot be reliably determined upfront? Are there empirical results for such edge cases?*
>
> **Response.** In our MulVuln approach, the shared cross-language knowledge is encoded by a pre-trained language model’s encoder (e.g., CodeT5’s encoder), while the parameter pool is specifically designed to model language-specific features, enabling the model to adapt to the unique characteristics of each programming language. By capturing both shared and language-specific knowledge, the proposed framework achieves more robust and effective multilingual vulnerability detection.
>
> Within the scope of multilingual vulnerability detection tasks, as also studied in prior work (e.g., Shu et al., 2025) and given the availability of multilingual datasets, each source code sample comes from a single, known programming language, *reflecting the typical scenario in practice*.
>
> Due to the lack of publicly available datasets containing mixed-language code or code where the language cannot be reliably determined upfront, and because this scenario is beyond the scope of the current study and previous work, we have not empirically evaluated such cases.
>
> Importantly, we note that our MulVuln approach, with the Parameter Selection via Key–Parameter Query mechanism, does not require the programming language to be known in advance. The model dynamically selects the most appropriate parameter from the parameter pool to capture the language-specific knowledge of each input solely based on the input embedding (via the [CLS] token representation). In principle, this mechanism could be applied to mixed-language or ambiguous code, and exploring such scenarios is an interesting direction for future work.
>
> **Q.** *Results currently lack confidence intervals/statistical significance analysis. Are the observed improvements in F1-score over the best PLM/LLM baselines robust to different seeds or test splits?*
>
> **Response.** In our experiments, consistent with prior work investigating a range of LLM-based approaches in multilingual vulnerability detection (e.g., Shu et al., 2025), we used the pre-defined training, validation, and test splits of the REEF dataset rather than creating alternative splits, ensuring a fair and objective evaluation.
>
> For the baselines, *we used the source code released by the authors, followed their recommended hyperparameters, and reported the best performance for each method*. Furthermore, for LLM-based baseline methods, in line with (e.g., Shu et al., 2025), *we explored diverse strategies, including zero-shot prompting, few-shot prompting, instruction fine-tuning with zero-shot, and instruction fine-tuning with few-shot settings, again reporting the best performance*. Therefore, although we did not perform multiple runs with different random seeds, using the pre-defined dataset splits and the recommended hyperparameters of the baselines ensures a fair and reproducible comparison.
>
> Across the evaluated metrics, our MulVuln approach achieves a much higher Recall and a substantially better overall balance, as reflected in its superior F1-score, indicating that the improvements are meaningful and robust.
>
> In short, *our experiments follow standard practice in software vulnerability detection (as done in related work)*, using well-established dataset splits and recommended hyperparameters of the baselines, ensuring reliable, reproducible, and fair comparisons.
>
> To further demonstrate the efficiency and stability of our MulVuln approach, as mentioned in our model configuration, MulVuln requires tuning a small set of hyperparameters, primarily the regularization weight $\lambda$. We evaluate $\lambda$ over the range {$\{1 \times 10^{-1}, 3 \times 10^{-1}, 1 \times 10^{-2}, 3 \times 10^{-2}\}$}. Across these settings, MulVuln exhibits very low variance (0.315) in F1-score. Achieving high performance, around 72.20\% in F1-score, together with this low variance, indicates that MulVuln’s performance is stable and consistent across reasonable hyperparameter choices, further supporting the robustness of the proposed method.

---

### Official Review · Reviewer_2csG · 2025-10-31

**Soundness:** 2
**Presentation:** 2
**Contribution:** 2
**Rating:** 4
**Confidence:** 4

**Summary:**

This paper proposes MulVuln, a novel approach for multilingual software vulnerability detection. The authors highlight that most existing AI-based detection methods are limited to a single programming language, which is insufficient for modern software systems that are often complex and written in multiple languages. MulVuln is designed to capture both shared knowledge that generalizes across different languages and language-specific knowledge that reflects unique coding conventions. The approach was evaluated on the real-world REEF dataset, which includes 4,466 CVEs across seven different programming languages. The experiments demonstrated that MulVuln significantly outperformed thirteen state-of-the-art baselines, achieving an F1-score improvement of 1.45% to 23.59%.

**Strengths:**

Important Problem: The paper correctly identifies and addresses a significant, practical gap in SVD research: the lack of effective, multilingual models for real-world codebases.

Clear Methodology: The proposed MulVuln approach is simple, intuitive, and clearly explained. The two selection mechanisms (instance-based query vs. language-aware training) are sensible explorations of the design space.

**Weaknesses:**

**Limited and Unclear Empirical Evaluation**: The experimental design suffers from two significant gaps regarding contemporary baselines: (1) While the paper states the use of various prompting strategies (zero-shot, few-shot, and instruction-based few-shot prompting), the final result in RQ1 is an aggregate, single score for all LLMs. (2) The paper overlooks several highly relevant and recently published baselines based on both PLMs [1-2] and LLMs [3-4], which significantly weakens the claim of achieving state-of-the-art performance.

[1] Distinguishing Look-Alike Innocent and Vulnerable Code by Subtle Semantic Representation Learning and Explanation

[2] SCALE: Constructing Structured Natural Language Comment Trees for Software Vulnerability Detection

[3] Boosting Vulnerability Detection of LLMs via Curriculum Preference Optimization with Synthetic Reasoning Data

[4] Collaboration to Repository-Level Vulnerability Detection


**Limited Scalability Discussion**: The model's design, especially Eq. 2, assumes a closed set of $S$ languages, with $S$ parameter matrices. This does not scale well to dozens of languages and offers no clear path for handling languages unseen during training (a critical aspect of true multilingual generalization).

**Questions:**

1. The authors state that "zero-shot, few-shot and instruction-based few-shot prompting were adopted for DeepSeek-Coder, Code Llama, Llama 3, GPT-3.5-Turbo and GPT-4o". However, the authors later claim that LoRA fine-tuning was also "applied" (l. 365). Please explicitly specify which of the above models were actually LoRA-fine-tuned and which were only prompt-engineered. Besides, RQ-1 reports a single bar per model; it is impossible to tell whether the number comes from zero-shot, few-shot or instruction-based few-shot. Clarify the exact prompting protocol used for each reported result.

2. Does LoRA fine-tuning of LLMs surpass the performance of MulVuln? Present a head-to-head comparison (MulVuln vs. LoRA-LLM) on the same test split so that the benefit of your adaptation strategy can be quantified.

3. If CodeT5-base already delivers strong results, have you experimented with larger checkpoints of the CodeT5+ family (e.g., 2B or 16B parameters)?

4. Can MulVuln generalise to unseen programming languages, or at least adapt from a handful of training samples in a new language? Report zero-/few-shot transfer results on at least one language never seen during training to validate the claim of language-agnostic vulnerability detection.

---

> ### Author Response · Authors · 2025-11-21
> **Authors’ responses to Reviewer 2csG (Part 1)**
>
> **We thank the reviewer for reviewing our paper and providing comments and questions for further enhancements and clarifications. Below, we provide responses to all the comments and questions.**
>
> **Q.** *The paper overlooks several highly relevant and recently published baselines based on both PLMs and LLMs, which significantly weakens the claim of achieving state-of-the-art performance.*
>
> **Response.** In our experiments, we compare our MulVuln approach against 13 strong, widely adopted, and recent baselines, including both pre-trained language models (PLMs) and large language models (LLMs) such as CodeBERT, GraphCodeBERT, LineVul, UniXcoder, CodeT5, CodeT5+, DeepSeek-Coder, Code Llama, Llama 3, GPT-3.5-Turbo, and GPT-4o. These models have demonstrated effectiveness in software vulnerability detection as well as the ability to handle multilingual vulnerability detection.
>
> We would like to highlight that the LLM-based methods included in our experiments were adapted for the task using diverse, effective approaches, as used in prior work. For example, CodeT5 and CodeT5+ were fine-tuned using their encoder components followed by a multi-layer feedforward classifier head, while models such as DeepSeek-Coder, Code Llama, Llama 3, GPT-3.5-Turbo, and GPT-4o were instruction fine-tuned in addition to prompt engineering. These adaptations enable the models to perform vulnerability detection effectively while leveraging their inherent ability to handle the multilingual setting.
>
> Although some baselines, as you mentioned, are not included, the selected 13 effective and state-of-the-art models provide a broad and representative evaluation of current approaches applied in multilingual vulnerability detection.
>
> **Q.** *The authors state that "zero-shot, few-shot and instruction-based few-shot prompting were adopted for DeepSeek-Coder, Code Llama, Llama 3, GPT-3.5-Turbo and GPT-4o". However, the authors later claim that LoRA fine-tuning was also "applied". Please explicitly specify which of the above models were actually LoRA-fine-tuned and which were only prompt-engineered.*
>
> **Response.** As presented in Section 4.4 (model’s configurations), in our experiments, in line with (Shu et al., 2025), for experiments with closed-source LLMs, we used GPT-3.5-Turbo (model version gpt-3.5-turbo-0125) and GPT-4o (model version gpt-4o-2024-08-06) through OpenAI’s API (OpenAI, 2024).
>
> For open-source LLMs, we utilized Hugging Face checkpoints for DeepSeek-Coder (6.7B parameters), Code Llama (7B parameters), and Llama 3 (8B parameters), and applied Low-Rank Adaptation (LoRA) during instruction fine-tuning.
>
> For these LLMs (i.e., GPT-3.5-Turbo, GPT-4o, DeepSeek-Coder, Code Llama, and Llama 3), we employed zero-shot prompting, few-shot prompting, instruction fine-tuning with zero-shot prompting, and instruction fine-tuning with few-shot prompting, following (Shu et al., 2025).
>
> *\*Regarding an additional comment related to the question, besides, RQ-1 reports a single bar per model; it is impossible to tell whether the number comes from zero-shot, few-shot or instruction-based few-shot.*
>
> **Response.** As also described in Section 4.4 (model’s configurations), for each of these methods, we report the best result regarding F1-score across the applied prompting and fine-tuning strategies. This explains why only a single bar per model is shown.

---

> ### Author Response · Authors · 2025-11-21
> **Authors’ responses to Reviewer 2csG (Part 2)**
>
> **Q.** *Does LoRA fine-tuning of LLMs surpass the performance of MulVuln? Present a head-to-head comparison (MulVuln vs. LoRA-LLM) on the same test split so that the benefit of your adaptation strategy can be quantified.*
>
> **Response.** In our experiments, to open-source LLM baselines, i.e., DeepSeek-Coder (6.7B parameters), Code Llama (7B parameters), and Llama 3 (8B parameters), we applied Low-Rank Adaptation (LoRA) during instruction fine-tuning.
>
> As shown in Table 1, our proposed MulVuln method significantly outperformed these models by a wide margin across all the used metrics, including F1-score, Recall, and Precision. Specifically, MulVuln achieves improvements ranging from 5.40% to 49.07% in Precision, 5.36% to 8.17% in Recall, and 7.94% to 23.59% in F1-score over these baselines.
>
> **Q.** *If CodeT5-base already delivers strong results, have you experimented with larger checkpoints of the CodeT5+ family (e.g., 2B or 16B parameters)?*
>
> **Response.** In our experiments, to demonstrate the effectiveness of our proposed approach, we used the base versions (220M) of CodeT5 and CodeT5+. This allows us to highlight the superiority of our method while maintaining computational efficiency.
>
> While larger checkpoints (e.g., 2B or 16B) could potentially improve performance, larger models do not always lead to proportional gains and come with substantially higher computational costs, making experiments less efficient and harder to reproduce. Our focus was to evaluate the effectiveness of MulVuln in a setting that is practical and reproducible, using standard model sizes commonly adopted in the field.
>
> **Q.** *Can MulVuln generalise to unseen programming languages, or at least adapt from a handful of training samples in a new language? Report zero-/few-shot transfer results on at least one language never seen during training to validate the claim of language-agnostic vulnerability detection.*
>
> **Response.** Our study is for multilingual vulnerability detection, where the goal is to leverage both shared and language-specific knowledge to improve detection performance across multiple programming languages.
>
> Evaluating generalization to completely unseen languages is not aligned with the problem statement (described in Section 3.1) and beyond the scope of our study. Regarding adaptation from a few training samples in a new language, the REEF dataset we use already includes languages with varying numbers of samples, ranging from hundreds (e.g., C#) to thousands (e.g., JavaScript), covering a diverse range of low-resource to high-resource scenarios.
>
> In our study, “language-agnostic” means that a model can operate on input code without needing to know which programming language it is written in. Our MulVuln approach, when using the Parameter Selection via Key–Parameter Query mechanism in the training phase, is language-agnostic, as it does not require the programming language to be known beforehand. Moreover, our approach does not require language identification at inference time. This does not imply generalization to completely unseen programming languages, which is beyond the scope of existing work on multilingual vulnerability detection. We will explore this aspect in future work.

---

### Official Review · Reviewer_zUau · 2025-11-03

**Soundness:** 2
**Presentation:** 2
**Contribution:** 2
**Rating:** 2
**Confidence:** 3

**Summary:**

This paper proposes MulVuln, a multilingual vulnerability detection approach that augments pre-trained language models with a learnable parameter pool to capture both shared and language-specific knowledge. The method selects appropriate parameters via key-based matching or language masking, concatenates them with input embeddings, and processes them through CodeT5's encoder. Experiments on the REEF dataset covering 7 programming languages show F1-score improvements of 1.45-2.81% over fine-tuned CodeT5/CodeT5+.

**Strengths:**

The paper addresses a important problem of multilingual vulnerability detection with a clear and intuitive approach. The experimental evaluation is comprehensive, covering 7 programming languages on the REEF dataset with 4,466 CVEs and comparing against 13 diverse baselines spanning deep learning models, pre-trained language models, and large language models. The proposed two-component design balances shared cross-language knowledge with language-specific features, and the visualizations provide useful insights into how the parameter pool operates. The writing is generally clear and the methodology is well-explained.

**Weaknesses:**

The most critical flaw is the absence of parameter-efficient fine-tuning baselines like Prefix-Tuning, LoRA, and Adapters, which are directly comparable to the proposed approach and essential for establishing novelty—without these comparisons, the contribution reduces to applying existing prefix-tuning techniques to vulnerability detection. All experimental results lack statistical rigor with single-run evaluations, no error bars, and no significance testing, making it impossible to determine whether the improvements are meaningful or simply noise. The claims about learning "language-specific knowledge" are inadequately supported, with no analysis of what the parameter pool actually encodes, no parameter similarity matrices across languages, and no cross-language transfer experiments to validate the separation of shared versus specific features. Critical ablation studies are missing, particularly varying the parameter pool size S beyond the default value of 7 and testing different query functions beyond the [CLS] token. The generalization capabilities remain completely untested through leave-one-language-out experiments, temporal splits, or cross-project evaluation, which is problematic given the dataset's severe imbalance that goes unaddressed. Several experimental results are unexplained, such as why DeepSeek-Coder with 6.7B parameters achieves only 48.61% F1 while the much smaller CodeT5 performs better, and why GPT-4o's precision (74.54%) vastly exceeds MulVuln's (57.51%). The theoretical justification is entirely absent, with no explanation for why prepending 5 learnable tokens should capture language-specific knowledge or why the particular loss formulation in Equation 3 is appropriate.

**Questions:**

see in the Weaknesses

---

> ### Author Response · Authors · 2025-11-21
> **Authors’ responses to Reviewer zUau (Part 1)**
>
> **We thank the reviewer for reviewing our paper and providing comments and questions for further enhancements and clarifications. Below, we provide responses to all the comments and questions.**
>
> **Q.** *The most critical flaw is the absence of parameter-efficient fine-tuning baselines like Prefix-Tuning, LoRA, and Adapters, which are directly comparable to the proposed approach and essential for establishing novelty.*
>
> **Response.** Prefix-Tuning, LoRA, and Adapters are parameter-efficient fine-tuning techniques designed to adapt large language models to downstream tasks with significantly reduced computational cost, while still often approaching the performance achieved by fully fine-tuning these models.
>
> In our experiments, we applied LoRA to open-source large models, including DeepSeek-Coder, Code Llama, and Llama 3, combined with instruction fine-tuning. As shown in Table 1, *compared with these baselines, our MulVuln approach demonstrates consistently superior performance across all evaluated metrics, Precision, Recall, and F1-score, by a substantial margin*. Specifically, MulVuln achieves improvements ranging from 5.40% to 49.07% in Precision, 5.36% to 8.17% in Recall, and 7.94% to 23.59% in F1-score.
>
> **Q.** *Critical ablation studies are missing, particularly varying the parameter pool size S beyond the default value of 7 and testing different query functions beyond the [CLS] token.*
>
> **Response.** As presented in Section 3.2.2, the pool size is intentionally set to match the number of programming languages in the dataset, *aligning with the pool’s purpose of representing per-language knowledge*. The main goal of the parameter pool is to encode the language-specific knowledge (representation) of each programming language; therefore, setting $S$ equal to the number of languages is the most appropriate and theoretically grounded choice, as stated in the methodology section. We explored varying the pool size in the ablation study (Appendix A.5.2), where the impact of different parameter pool sizes is evaluated.
>
> Using the [CLS] token as the query vector is a standard and effective approach, as it represents the overall embedding of the input sequence. Alternative query strategies, such as averaging or summing token embeddings, would introduce additional complexity without clear benefits. Thus, the [CLS] token offers a simple and robust choice consistent with best practices in transformer-based classification.
>
> **Q.** *Why DeepSeek-Coder (6.7B) achieves only 48.61% F1 while the much smaller CodeT5 performs better.*
>
> **Response.** A larger parameter count does not necessarily lead to better performance, particularly when comparing models differing both in architecture and pretraining objectives.
>
> DeepSeek-Coder is a transformer-decoder-based model optimized for generation tasks (with the use of transformer-decoder, which often emphasizes pattern reproduction rather than semantic understanding), whereas CodeT5 uses an encoder–decoder architecture that is more effective for data understanding through its encoder component, making it better suited for vulnerability detection. In our study, we utilize the encoder component of CodeT5 followed by a multi-layer feedforward classifier head, a standard approach for adapting such models to classification tasks.
>
> **Q.** *Why GPT-4o’s precision exceeds MulVuln’s.*
>
> **Response.** The observed difference in precision between GPT-4o (74.54%) and MulVuln (57.51%) reflects the inherent trade-off between precision and recall. Models with higher precision tend to be more conservative, making fewer positive (i.e., vulnerable) predictions and thus fewer false positives, often at the expense of recall. Our MulVuln approach, in contrast, achieves much higher recall (MulVuln’s 96.96% vs. GPT-4o’s 67.22%) and a substantially better overall balance, as indicated by its superior F1-score (MulVuln’s 72.20% vs GPT-4o’s 70.69%). Since the F1-score is the harmonic mean of precision and recall, it provides a more reliable measure of overall performance across both metrics.

---

> ### Author Response · Authors · 2025-11-21
> **Authors’ responses to Reviewer zUau (Part 2)**
>
> **Q.** *The claims about learning "language-specific knowledge" are inadequately supported, with no analysis of what the parameter pool actually encodes.*
>
> **Response.** As presented in Section 3.2.2 (parameter pool for language-specific knowledge), the parameter pool is designed to encode language-specific knowledge (representations) through two mechanisms, including Parameter Selection via Key–Parameter Query and Language-Aware Parameter Masking.
>
> The Key--Parameter Query mechanism associates each parameter $P\_i$ in the parameter pool $\mathcal{P}$ with a learnable key $k\_i$, forming a collection of key–parameter pairs that encode language-specific knowledge. For each input $X$, the model computes a query vector $q(X)$ from the [CLS] representation of the input and selects the most relevant parameter $P\_X$ by comparing the query with all keys (e.g., via cosine similarity). This allows the model to dynamically and adaptively choose the parameter that best matches the input content.
>
> The Language-Aware Parameter Masking mechanism introduces supervision during training by restricting the selection to the parameter associated with the input's language. Each language $\ell$ is assigned a fixed parameter index $i\_\ell$, and the query is restricted to selecting only from this language-specific parameter. While this guides the model during training, the parameter $P\_i$ and key representations $k\_i$ are still learned. Consequently, at inference time, the model can select the appropriate parameter for each input dynamically, making our MulVuln approach language-agnostic.
>
> Overall, these mechanisms enable the parameter pool to learn and retrieve language-specific representations in a structured manner. The pool acts as a memory of programming-language characteristics, allowing the model to apply the most suitable language-specific parameter for each input.
>
> Furthermore, our ablation study (Appendix A.2.2) offers a direct examination of the parameter pool and query distributions, demonstrating that the pool serves as a consistent reference for language-specific knowledge (representations).
>
> **Q.** *No parameter similarity matrices across languages.*
>
> **Response.** As presented in Section 3.2 (Methodology), in our proposed MulVuln approach, the shared cross-language knowledge is encoded by a pre-trained language model’s encoder (e.g., CodeT5’s encoder), while the parameter pool is specifically designed to model language-specific features, enabling the model to adapt to the unique characteristics of each programming language.
>
> By capturing both shared and language-specific knowledge, the proposed framework achieves more robust and effective multilingual vulnerability detection. This is supported by the rigorous experiments conducted on diverse real-world multilingual source code data (Table 1).
>
> **Q.** *The theoretical justification is absent, with no explanation for why prepending 5 learnable tokens should capture language-specific knowledge or why the loss formulation in Eq. (3) is appropriate.*
>
> **Response.** As described in Section 3.2.2, the parameter pool is designed to encode language-specific knowledge, enabling the model to adapt to the characteristics of each programming language. The parameter length $L\_p$ represents the number of learnable embedding vectors allocated to each language, defining the capacity of the parameter to encode language-specific knowledge (consistent with the token embedding dimension $D$). The rationale and impact of varying $L\_p$ were examined in the ablation study (Appendix A.5.1).
>
> As presented in Section 3.2.3, the overall training objective function (in Eq. (3)) combines two goals: (i) performing the main task effectively by detecting vulnerabilities accurately using both shared and language-specific representations via the first term (i.e., $\mathcal{L}\_{\text{CE}}\big(g\_\beta(f\_{plm\_{mha}}(X\_p)), Y\big)$), and (ii) maintaining meaningful parameter selection by ensuring that the input’s learned query feature aligns closely with the correct language-specific parameter key via the second term (i.e., $- \lambda \, \phi(q(X), k\_{i^*}$). This alignment enables the model to learn a stable mapping between inputs and their most relevant parameters, allowing it to dynamically select the most appropriate language-specific parameter for each input X. The selected parameters are then combined with the input embeddings in the PLM’s multihead-attention layers, allowing the model to integrate shared knowledge captured by the PLM with enhanced language-specific information, supporting more effective multilingual vulnerability detection.

---

> ### Author Response · Authors · 2025-11-21
> **Authors’ responses to Reviewer zUau (Part 3)**
>
> **Q.** *The generalization capabilities remain untested through leave-one-language-out experiments, temporal splits, or cross-project evaluation, which is problematic given the dataset's severe imbalance that goes unaddressed.*
>
> **Response.** We acknowledge the reviewer’s concern regarding leave-one-language-out experiments. While such evaluations could provide additional insights, they are not aligned with the problem statement (described in Section 3.1) and are beyond the scope of this study. Our study is for multilingual vulnerability detection, where the goal is to leverage both shared and language-specific knowledge to improve detection performance across multiple programming languages.
>
> In terms of *generalization and realistic evaluation*, the REEF dataset used contains 4,466 CVEs with 30,987 patches across seven programming languages, collected from real-world vulnerabilities in the National Vulnerability Database (NVD) and Mend’s CVE list, spanning 2016–2023. This diversity in programming languages, code patterns, and temporal coverage provides a natural test of the model’s ability to generalize to diverse and previously held-out code samples, reflecting realistic multilingual vulnerability detection scenarios.
>
> To ensure fair and direct comparison with the baselines applied in the multilingual setting, we follow the established REEF dataset splits (as in Shu et al., 2025) and report averaged performance across all seven programming languages for each method. This aggregated evaluation reflects the overall effectiveness of each model in multilingual vulnerability detection, emphasizing generalization across languages rather than optimization for a single language individually.
>
> We report Recall, Precision, and F1-score for all methods in Table 1, ensuring that the evaluation reflects performance on both vulnerable and non-vulnerable classes, mitigating the effect of class imbalance in the results.
>
> **Q.** *All experimental results lack statistical rigor with single-run evaluations, no error bars, and no significance testing, making it impossible to determine whether the improvements are meaningful or simply noise.*
>
> **Response.** In our experiments, consistent with prior work investigating a range of LLM-based approaches in multilingual vulnerability detection (e.g., Shu et al., 2025), we used the pre-defined training, validation, and test splits of the REEF dataset rather than creating alternative splits, ensuring a fair and objective evaluation.
>
> For the baselines, *we used the source code released by the authors, followed their recommended hyperparameters, and reported the best performance for each method*. Furthermore, for LLM-based baseline methods, in line with (e.g., Shu et al., 2025), *we explored diverse strategies, including zero-shot prompting, few-shot prompting, instruction fine-tuning with zero-shot, and instruction fine-tuning with few-shot settings, again reporting the best performance*. Therefore, although we did not perform multiple runs with different random seeds, using the pre-defined dataset splits and the recommended hyperparameters of the baselines ensures a fair and reproducible comparison.
>
> Across the evaluated metrics, our proposed MulVuln achieves a much higher Recall and a substantially better overall balance, as reflected in its superior F1-score, indicating that the improvements are meaningful and robust.
> In short, our experiments follow standard practice in software vulnerability detection (as done in related work), using well-established dataset splits and recommended hyperparameters of the baselines, ensuring reliable, reproducible, and fair comparisons.
>
> To further demonstrate the efficiency and stability of our MulVuln approach, as mentioned in the model configuration (Section 4.4), MulVuln requires tuning a small set of hyperparameters, primarily the regularization weight $\lambda$. We evaluate $\lambda$ over the range {$\{1 \times 10^{-1}, 3 \times 10^{-1}, 1 \times 10^{-2}, 3 \times 10^{-2}\}$}. Across these settings, MulVuln exhibits very low variance (0.315) in F1-score. Achieving high performance, around 72.20\% in F1-score, together with this low variance, indicates that MulVuln’s performance is stable and consistent across reasonable hyperparameter choices, further supporting the robustness of our proposed method.

---

### Meta-Review · Area_Chair_VC2x · 2026-01-06

**Summary:**

The reviewers generally agree that the paper addresses a relevant problem and proposes a clear and intuitive method for multilingual vulnerability detection. However, the suggested decision is mainly informed by consistent concerns across reviews regarding limited empirical rigor and unclear contribution, including missing or insufficiently strong baselines, lack of statistical validation, limited ablation and analysis, and weak empirical support for the claims about language-specific modeling and generalization.

**Reviewer Concerns:**

In the rebuttal, the authors clarified parts of the experimental setup and justified several design choices, which helps resolve some confusion about baselines and methodology. However, the core concerns remain largely outstanding: the experimental evidence is still based on single-run results without statistical support, the evaluation scope is limited to a single dataset and setting, and the paper does not yet provide strong empirical validation for its main conceptual claims or clearly establish novelty over existing parameter-efficient adaptation methods.

**Reviewer Scores:**

Overall, I believe most reviewers would tend to maintain their original scores after discussion. While some clarification may lead to slight positive adjustments, the main weaknesses identified in the initial reviews are not fundamentally resolved, and the overall score distribution would likely remain below the acceptance threshold.

---

### Decision · Program_Chairs · 2026-01-26

Reject